# MEMORY-ORCHESTRATED MULTI-PROMPT LEARNING FOR INFRARED AND VISIBLE IMAGE FUSION

## ABSTRACT

Infrared and visible image fusion aims to integrate complementary information from different modalities into a unified representation. However, existing methods lack the capability to leverage historical fusion experiences and generate modality-specific semantic guidance, thereby limiting their adaptability and fusion quality. To address these challenges, this study proposes a Memory-Orchestrated Multi-Prompt Learning Network that transforms fusion from a static feature combination process into a dynamic prompt-guided learning paradigm. Our method encompasses two core mechanisms: 1) Memory-driven experiential prompts that capture and reuse successful fusion patterns from historical cases through a CLIP-evaluated dynamic memory bank; 2) Graph-driven modality-specific prompts that model cross-modal semantic relationships via specialized semantic graph networks to generate targeted guidance for each modality. These dual prompts are jointly modulated across multiple scales and progressively integrated into the fusion process, enabling stable, interpretable, and transferable guidance for fusion decisions without relying on full supervision. Furthermore, we exploit residual priors to assess the salient complementarity of source features, thereby constraining the solution space and enhancing the model's effective perception of complementary characteristics. Extensive experiments, including both statistical metrics and performance on high-level vision tasks, demonstrate the effectiveness of the proposed method.

## 1 INTRODUCTION

Infrared and visible image fusion (IVIF) represents a critical image enhancement technique that integrates thermal radiation information from infrared images with textural details from visible images, generating more informative unified representations (Xu et al., 2020; Zhang & Demiris, 2023; Liu et al., 2024b). As a foundational vision task, IVIF significantly enhances the performance of downstream high-level vision tasks, including object detection (Liu et al., 2025a), scene analysis (Zheng et al., 2025), and autonomous navigation (Liu et al., 2023), by providing enriched multi-modal information.

The advancement of deep learning (DL) has provided powerful technical foundations for IVIF, where adaptive feature extraction and integration capabilities of deep networks effectively alleviate the limitations of hand-crafted rules inherent in traditional methods. Consequently, DL-based methods have become the predominant research paradigm. However, in the absence of ground truth supervision, existing DL-based methods typically rely on structural or attribute priors of source features to construct learning strategies that drive models to capture explicit cross-modal feature representations (Zhao et al., 2024a; 2023; 2025; Cheng et al., 2025). While effective to some extent, such constraints based on fixed loss functions or single priors struggle to provide stable guidance for generating high-quality fusion results. The fundamental challenge lies in translating the subjective notion of *'perceptual quality'* into learnable optimization objectives under unsupervised conditions. Recent research efforts have attempted to bridge fusion processes with high-level vision tasks by establishing *'fusion-task'* connections, injecting task semantics into fusion procedures to enhance model expressiveness (Liu et al., 2025a; Chen et al., 2025; Wu et al., 2025). However, the weak supervision nature of task semantics limits their generalizability, resulting in constrained performance when facing unknown tasks. Inspired by the rapid development of prompt learning, some researchers have leveraged vision-language models to guide IVIF models in learning gener-

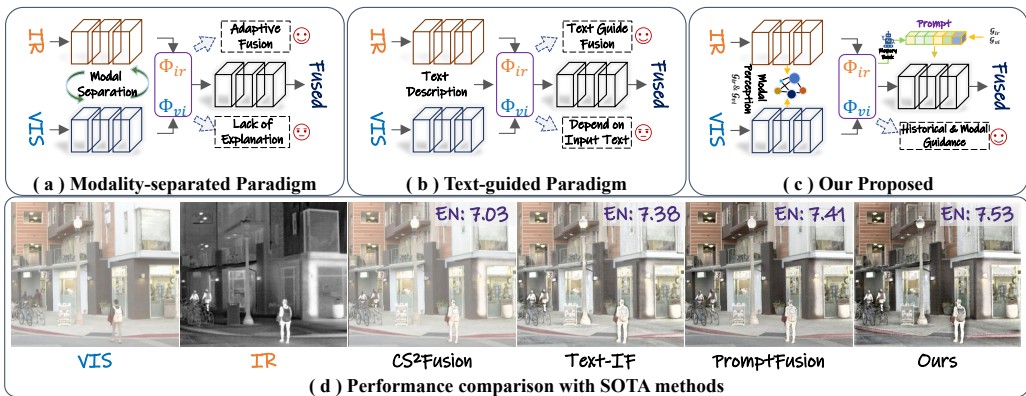

Figure 1: (a)-(c) present a comparative analysis of different fusion paradigms, including modality-specific modeling, text-guided fusion, and our proposed. (d) demonstrates the superior effectiveness of our proposed M2PN through comparisons with SOTA methods: modality-specific-based CS²Fusion (Wang et al., 2024), text-guided fusion Text-IF (Yi et al., 2024) and PromptFusion (Liu et al., 2024a).

alizable representations, thereby promoting high-quality image generation (Zhang et al., 2025; Li et al., 2025a; Liu et al., 2024a). Particularly in ground-truth-free scenarios, the integration of textual descriptions endows models with enhanced feature perception capabilities, enabling them to make informed fusion decisions based on semantic guidance rather than blind feature combination (Yi et al., 2024). Despite these advances, prompt learning-based IVIF methods face several critical challenges: 1) Existing methods primarily rely on explicitly modeled prompts and lack the ability to learn from historical successful fusion cases, failing to generate effective experiential prompts for current fusion tasks; 2) The distinct characteristics of infrared and visible modalities necessitate specialized semantic guidance, yet current methods fail to generate modality-specific prompts that account for inherent modal differences; 3) In the absence of ground truths and under cross-modal distribution inconsistency, translating human-perceived quality into learnable constraints for stable fusion quality remains unresolved.

Based on the above findings, this work proposes a Memory-Orchestrated Multi-Prompt Learning Network (M2PN) that transforms the fusion process into a dynamic, prompt-guided learning paradigm through adaptive prompt generation. Unlike existing paradigms that rely on explicit CLIP guidance, our method leverages CLIP's robust evaluation capabilities to construct a self-evolving Dynamic Memory Bank (DMB) that stores high-quality fusion feature representations from historical learning episodes. The model subsequently queries this memory bank to capture and reuse successful fusion patterns, generating experiential prompts to guide current fusion decisions. Additionally, we design a Cross-Modal Semantic Graph Network (CSGN) that models modality-specific semantic relationships between infrared and visible images. Through modality-specialized graph representation learning, CSGN generates unique semantic guidance prompts for each modality. The experiential and modality-specific prompts are jointly modulated across multiple scales, with adaptive prompt weight adjustment based on feature responses, and progressively injected to guide fusion image generation. Furthermore, we leverage the structural priors of the residual maps to evaluate the complementary features of source features, employing a weighted loss function to constrain the solution space and enhance the model's effective perception of complementary features. Extensive experimental results demonstrate that memory-guided multi-prompts learning can effectively guide the model in leveraging complementary contextual aggregation, achieving more competitive performance compared to SOTA methods. The main contributions of this work are summarized as follows:

- We propose M2PN, which transforms fusion from a static feature combination process into a dynamic prompt-guided learning paradigm.

- We introduce two complementary prompt generation strategies, memory-driven experiential prompts that leverage CLIP-evaluated historical fusion experiences through dynamic

retrieval, and graph-driven modality-specific prompts that generate specialized knowledge through semantic information propagation and aggregation in graph structures.

- Efficient modules, such as memory-guided fusion and residual-weighted map mechanisms that effectively enhance M2PN's performance through progressive prompt injection and complementarity-aware feature learning.

## 2 RELATED WORK

**DL-based IVIF.** Deep learning for IVIF has evolved along several interconnected threads. Early methods emphasized preserving complementary structural and visual cues from source images via tailored architectures and priors-driven objectives (Wang et al., 2024; 2025b; Zheng et al., 2025). Within this paradigm, CNN-based frameworks, DenseFuse (Li & Wu, 2018), U2Fusion (Xu et al., 2020), and FusionGAN (Ma et al., 2019), established foundational pipelines that assess input importance to retain salient source features. However, their local receptive fields inherently constrain long-range dependency modeling and cross-modal interaction (Zhao et al., 2023; Liu et al., 2025a). To overcome these limits, transformer-based approaches leverage self-attention to capture global context and facilitate richer cross-modal interactions. Representative works such as SwinFusion (Ma et al., 2022), CDDFuse (Zhao et al., 2023), and YDTR (Tang et al., 2022b) demonstrate that long-range spatial relationships between IR and VIS modalities can be explicitly modeled, leading to more robust fusion strategies. Building further, diffusion-based models introduce generative priors and iterative denoising to encode distributions of source features. Dif-Fusion (Yue et al., 2023) pioneers this direction by casting channel distribution construction as a diffusion process, while DSPFusion (Tang et al., 2025) and DRMF (Tang et al., 2024) exploit diffusion's stochastic sampling to enhance degradation resistance under challenging conditions. In parallel, task-oriented fusion integrates feedback from downstream vision tasks to guide optimization. TarDAL (Liu et al., 2023) jointly optimizes fusion and detection, and DCEvo (Liu et al., 2025a) employs evolutionary learning to balance multi-objective trade-offs. Yet, despite clear gains under matched settings, such pipelines may generalize poorly to unknown or shifting downstream tasks, highlighting the need for experience-aware and task-agnostic guidance.

**Memory Mechanisms.** Orthogonal to the choice of backbone, memory mechanisms endow feed-forward models with the capacity to store, retrieve, and reuse informative representations across instances, thereby compensating for the myopic nature of one-shot processing (Liu et al., 2025b; Zhou et al., 2024a;b). In contrastive learning, MoCo (He et al., 2020) stabilizes negative sampling through a momentum-updated memory bank, improving representation consistency at scale. For video object segmentation, QDMN (Liu et al., 2025b) introduces quality-guided updates so that high-quality frames are preferentially retained, reinforcing temporal coherence. Related ideas appear in person re-identification, where adaptive memories continually refine identity prototypes from mini-batch instances (Yin et al., 2023), and in video-text retrieval, where memory banks help maintain temporal correspondences across modalities to support robust cross-modal alignment (Wang et al., 2022). Collectively, these results suggest that explicit memory can accumulate experiential knowledge beneficial for dynamic, context-dependent tasks—an ability also desirable for IVIF.

**Prompt Learning.** Concurrently, prompt learning offers a complementary route to adapt pre-trained models with minimal overhead by injecting contextual signals (Khattak et al., 2023; Ma et al., 2023; Liao et al., 2025; Zhang et al., 2024). Built on CLIP (Radford et al., 2021), vision–language prompts have been shown to transfer semantic priors effectively across tasks such as detection (Ma et al., 2023), style transfer (Kwon & Ye, 2022), and image enhancement (Liang et al., 2023), often surpassing traditional unsupervised cues by operating within semantically grounded latent spaces (Zhou et al., 2022). Motivated by these advances, IVIF studies have begun to incorporate textual guidance: IF-FILM (Zhao et al., 2024b) extracts explicit text cues from source images to steer fusion, Prompt-Fusion (Liu et al., 2024a) uses vision–language models to refine object-aware interactions, and Text-IF (Yi et al., 2024) leverages textual priors to break ground-truth bottlenecks for degradation-aware and interactive fusion. Despite these encouraging steps, current prompt-based IVIF faces three coupled limitations: (i) reliance on handcrafted or pre-defined prompts, which constrains adaptability; (ii) the absence of mechanisms to accumulate and reuse successful fusion experiences as compact guidance; and (iii) modality-agnostic prompt generation that overlooks distinct IR/VIS characteristics. These gaps motivate our objective: to automatically derive experiential, modality-aware

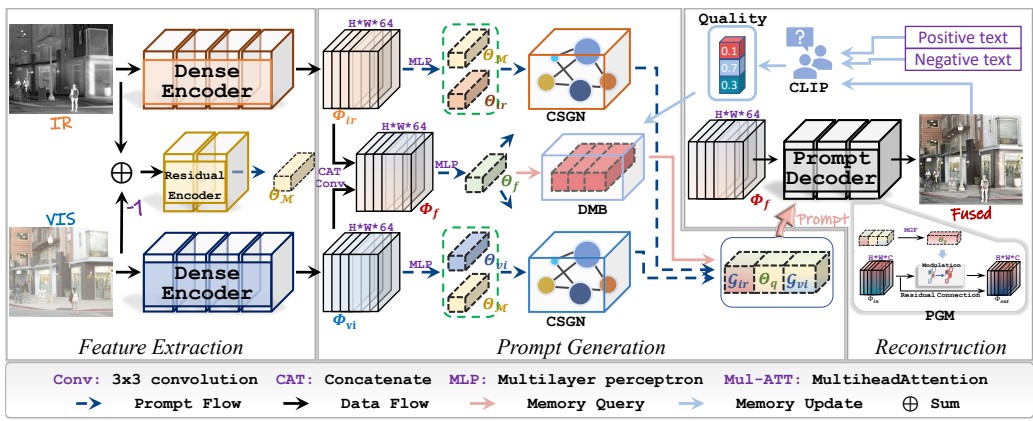

Figure 2: The framework of our memory-orchestrated multi-prompt learning network.

prompts that guide fusion without explicit supervision, thereby combining the strengths of memory and prompting in a unified framework.

## 3 METHODOLOGY

Contemporary IVIF methods are fundamentally limited by treating each fusion instance as an isolated optimization problem, thereby discarding valuable knowledge from successful fusion experiences and applying uniform processing strategies that neglect the inherent heterogeneity between infrared and visible modalities. In contrast, human visual perception demonstrates superior fusion capabilities by unconsciously leveraging experiential knowledge from previous scenarios while naturally adapting processing strategies to honor each modality's distinctive characteristics. This cognitive mechanism inspires our dual-prompt learning framework, which transforms the fusion paradigm from $fused = \mathbb{N}(IR, VIS; \Psi)$ to $fused = \mathbb{N}(IR, VIS, \mathbb{B}, \mathcal{G}; \Psi)$, where $fused$, $IR$, and $VIS$ represent the fused image, infrared image, and visible image, respectively. $\mathbb{N}$ represents the fusion function with static parameters $\Psi$. $\mathbb{B}$ encapsulates accumulated experiential knowledge from historical successful cases, and $\mathcal{G}$ captures modality-specific semantic understanding through graph-based cross-modal reasoning. The semantic component $\mathcal{G}$ is operationalized through a GNN that models cross-modal relationships as learnable node interactions rather than static feature combinations. This graph structure enables flexible information propagation along semantically meaningful pathways, capturing complex interdependencies between IR and VIS modalities that conventional operations cannot adequately represent. The framework thus addresses both temporal learning through $\mathbb{B}$ and structural reasoning through $\mathcal{G}$, enabling adaptive fusion decisions guided by accumulated experience and cross-modal semantic understanding. As illustrated in Figure 2, our M2PN operates through a three-stage pipeline: feature extraction, prompt generation, and reconstruction. This architecture transforms traditional static fusion into a dynamic, prompt-driven learning paradigm.

### 3.1 FEATURE EXTRACTION

We employ a Siamese-DenseEncoder (Wang et al., 2024) architecture to extract complementary feature representations $\Phi_{ir}$ and $\Phi_{vi}$ from $IR$ and $VIS$, respectively. The DenseEncoder leverages dense connectivity patterns to capture multi-scale feature hierarchies while preserving fine-grained details across different semantic levels. Additionally, we introduce a lightweight residual encoder composed of two convolutional layers to extract residual features $\Phi_{\mathcal{M}}$ from the residual map $\mathcal{M} := IR - VIS$, which captures the fundamental modality differences and provides a structural prior for complementarity perception (Wang et al., 2025a; He et al., 2023; Zheng et al., 2025).

### 3.2 PROMPT GENERATION

The extracted source features $\Phi_{ir}$ and $\Phi_{vi}$ are concatenated to generate an initial fused representation $\Phi_f$, which serves as the foundation for subsequent processing. (i) It acts as the core features

for fused image reconstruction; (ii) It collaborates with the residual feature $\Phi_{\mathcal{M}}$ and the source features to construct graph architectures, which are fed into the CSGN to generate modality-specific prompts through cross-modal semantic learning; (iii) It functions as a query mechanism to retrieve historical representations from the DMB, facilitating the generation of experiential prompts based on successful fusion patterns.

**Cross-Modal Semantic Graph Network (CSGN).**
To generate modality-specific prompts that capture the intrinsic characteristics of each modality, we design a CSGN to model semantic relationships through structured graph representations, as illustrated in Figure 3. Specifically, for each modality (IR and VIS), we construct a three-node semantic graph $\mathcal{G} = (\mathcal{V}, \mathcal{E})$, where the node set $\mathcal{V} = \{\Theta_{ir}/\Theta_{vi}, \Theta_f, \Theta_{\mathcal{M}}\}$. Each node's representation is derived through global average pooling followed by linear projection: $h_i = Proj(GAP(\Phi_i))$, where $i \in \{\Theta_{ir}/\Theta_{vi}, \Phi_f, \Phi_{\mathcal{M}}\}$.

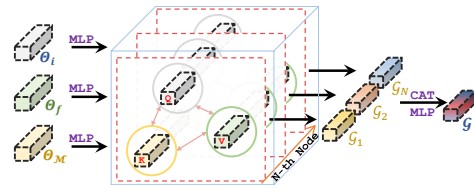

Figure 3: Pipeline for the MGF.

The graph employs multi-head cross-modal attention mechanisms to enable semantic information propagation across nodes:

$$Att(Q, K, V) = Softmax\left(\frac{CAT(Q_n, K_n) \cdot W_{attn}}{\sqrt{d_k}}\right) V_n \qquad (1)$$

where $Q_n$, $K_n$, and $V_n$ represent the *query*, *key*, and *value* projections for the $n$-th attention head, respectively. The CSGN processes $IR$ and $VIS$ modalities independently through dedicated attention layers, generating modality-specific graph representations $\mathcal{G}_{ir}$ and $\mathcal{G}_{vi}$ that encapsulate specialized semantic guidance for each modality.

**Dynamic Memory Bank (DMB).** To leverage historical fusion experiences, we implement a learnable memory mechanism that stores and retrieves high-quality fusion patterns. The DMB maintains a memory matrix $\mathcal{M} \in \mathbb{R}^{N \times D}$, where $N$ represents the memory capacity and $D$ denotes the feature dimensionality. The memory bank operates through three sequential processes: similarity-based retrieval, quality evaluation, and dynamic updating.

Given the current fused features $\Phi_f$, we first extract a global representation $\Theta_f = Proj(GAP(\Phi_f))$ and compute cosine similarities with stored memory entries:

$$s_i = \frac{\Theta_f \cdot \mathbb{M}_i}{||\Theta_f|| \cdot ||\mathbb{M}_i||} \qquad (2)$$

The experiential prompt is generated through weighted aggregation: $\Theta_q = \sum_{i=1}^{N} \alpha_i \mathbb{M}_i$, where $\alpha_i = Softmax(s_i)$.

For quality assessment, we employ a CLIP-based evaluator that addresses the challenge of defining fusion quality without ground truth supervision. Rather than direct textual constraints on fusion generation, which suffers from semantic ambiguity and feature mismatch issues, we leverage CLIP's evaluation capability on well-defined quality attributes (texture, contrast, brightness). The quality score is computed as:

$$Q_{CLIP} = \delta(Sim(I_f, T_{pos}) - Sim(I_f, T_{neg})) \qquad (3)$$

where $\delta$ denotes the sigmoid function, $I_f$ represents the CLIP encoding of the fused image, $Sim(x, y)$ calculates the cosine similarity of $x$ and $y$, and $T_{pos}$, $T_{neg}$ represent positive and negative quality descriptions, respectively.

The memory bank employs adaptive thresholding to selectively store high-quality experiences. The threshold $\tau_t$ is dynamically adjusted based on historical quality distributions:

$$\tau_t = \mu_{hist} + \kappa \cdot \sigma_{hist} \qquad (4)$$

where $\mu_{hist}$ and $\sigma_{hist}$ represent the historical mean and standard deviation of quality scores, and $\kappa$ is a learnable scaling parameter. Only fusion instances satisfying $Q_{CLIP} > \tau_t$ are incorporated into

the memory bank through momentum-based updates:

$$\mathbb{M}_j^{(t+1)} = (1 - \beta) \cdot \mathbb{M}_j^{(t)} + \beta \cdot \Theta_f \tag{5}$$

where $\beta$ control update rate is set to 0.1 and $j$ denotes the memory slot with highest similarity.

## 3.3 RECONSTRUCTION

The decoder adopts a stack of three Prompt Guidance Modules (PGMs). Each PGM consumes the current decoder features together with a dual–prompt design and returns a refined representation. Within each PGM, we couple Memory-Guided Fusion (MGF) with Adaptive Instance Normalization (AdaIN) (Huang & Belongie, 2017) to realize prompt-conditioned reconstruction. A residual connection preserves the input signal while enabling prompt-driven enhancement:

$$\Phi_{out} = AdaIN(\Phi_{in}, \Theta_p^i) + \Phi_{in}, \tag{6}$$

where $\Phi_{in}$ denotes the input features to a PGM and $\Theta_p$ is the fused prompt.

**Memory-Guided Fusion (MGF).** The fused prompt is produced by querying a modality bank with a memory embedding:

$$\Theta_p = MGF(\mathcal{G}_{ir}, \mathcal{G}_{vi}, \Theta_q), \tag{7}$$

Concretely, MGF implements a multi-head attention operator that uses $\Theta_q$ as the *query* and treats $[\mathcal{G}_{ir}, \mathcal{G}_{vi}]$ as *keys/values*, yielding memory-informed selection weights over the modality bank:

$$\alpha = \text{softmax}\left( \frac{(\Theta_q W_q)\left([\mathcal{G}_{ir}, \mathcal{G}_{vi}] W_k\right)^\top}{\sqrt{d}} \right), \quad \Theta_p = \alpha\left([\mathcal{G}_{ir}, \mathcal{G}_{vi}] W_v\right), \tag{8}$$

where $W_q, W_k, W_v$ are learnable projections, $[\cdot, \cdot]$ denotes concatenation, $d$ is the head dimension, and $\alpha \in \mathbb{R}^{1 \times 2}$ encodes the memory-guided preference over $IR/VIS$ cues. This design jointly determines *'what to fuse'* (via MGF) and *'how to modulate'* (via AdaIN), while the residual connection preserves fidelity.

## 3.4 OBJECT FUNCTION

The training objective of our M2PN comprises two complementary components: a fusion loss $\mathcal{L}_f$ that guides the model to integrate cross-modal complementary features, and a modal separation loss $\mathcal{L}_{ctr}$ that enforces modality-specific prompt specialization through contrastive learning. The overall loss function is formulated as:

$$\mathcal{L}_{total} = \mathcal{L}_f + \mathcal{L}_{ctr} \tag{9}$$

**Fusion loss $\mathcal{L}_f$.** The fusion loss $\mathcal{L}_f$ consists of a weighted fidelity term $\mathcal{L}_w$ and a texture-structure preservation term $\mathcal{L}_s$, defined as:

$$\mathcal{L}_f = \mathcal{L}_w + \lambda \mathcal{L}_s \tag{10}$$

where $\lambda$ represent the trade-off factor. The weighted fidelity term constrains the solution space by introducing adaptive weighting mechanisms that drive the model to effectively preserve critical information from both modalities. To ensure that the fused image simultaneously maintains thermal target sensitivity from infrared images and detail richness from visible images, we formulate the fusion goal as an energy minimization framework with adaptive weight allocation based on quantified information contribution from each modality:

1) We design a dual-level saliency computation mechanism. The process first emphasizes temperature salient regions through global standardization, then integrates local contrast enhancement with intensity weighting to ensure thermal target regions receive higher saliency weights:

$$S_{ir} = \sigma(\hat{\text{IR}} + C(IR)) \cdot \sigma(\hat{IR}) \tag{11}$$

where $\hat{IR} = \frac{IR - \mu(IR)}{\sigma(IR)}$ represents the globally normalized infrared image, and $C(\cdot)$ denotes local window convolution for contrast enhancement.

2) To quantify the information complementarity between modalities, we introduce residual entropy analysis to assess the importance of modal differences. The entropy of the residual probability distribution is computed as:

$$H_R = -\mathbb{E}[p_R \log(p_R + \epsilon) + (1 - p_R) \log(1 - p_R)] \tag{12}$$

where $p_R = \sigma(\mathcal{M})$ represents the normalized residual probability. The complementarity weight $\lambda_c = \sigma(H_R)$ adaptively regulates the contribution of residual information based on modal consistency.

3) Based on information-theoretic principles, we transform the contribution degree of each modality into energy functions, where lower energy indicates superior information preservation:

$$E_{ir} = -(S_{ir} + \lambda_c \cdot p_R \cdot S_{ir}) \qquad E_{vi} = -((1 - S_{ir}) + \lambda_c \cdot p_R \cdot (1 - S_{ir})) \tag{13}$$

These energy functions simultaneously encode intrinsic saliency and complementary information, ensuring thermal target regions favor infrared contributions while texture-rich areas preserve visible light details. The pixel-level decision is achieved by comparing energy differences:

$$w_{ir} = \mathbb{I}(E_{ir} - E_{vi} < k \cdot \sigma(E_{ir} - E_{vi})) \tag{14}$$

where $\mathbb{I}$ is the indicator function and $k$ serves as a control parameter to ensure the robustness of the decision-making of $k \cdot \sigma(E_{ir} - E_{vi})$ under different scenarios. The complementary weight is computed as $w_{vi} = 1 - w_{ir}$.

Finally, the weighted fidelity loss is formulated as:

$$\mathcal{L}_w = \|w_{ir} \cdot IR - w_{ir} \cdot fused\|_1 + \|w_{vi} \cdot VIS - w_{vi} \cdot fused\|_1 \tag{15}$$

Moreover, to enhance visual quality and preserve structural details while avoiding common detail loss during fusion, we introduce a quality term combining structural similarity and texture preservation:

$$\mathcal{L}_s = \mathcal{SSIM}(fused, IR) + \mathcal{SSIM}(fused, VIS) + \|\nabla fused - Max(\nabla IR, \nabla VIS)\|_1 \tag{16}$$

where $\mathcal{SSIM}(\cdot)$ measures structural similarity, $Max(\cdot)$ and $\nabla(\cdot)$ represents the max function and the gradient operator, respectively.

**Modal separation loss $\mathcal{L}_{ctr}$.** To enhance the discriminability of modality-specific cues and ensure appropriate cue specialization, we follow Wang et al. (2024) to introduce a modality contrastive learning framework that aims to promote intra-modality consistency while enforcing semantic separation between modalities:

$$\mathcal{L}_{ctr} = \mathcal{L}_{min}(\mathcal{G}_{ir}, \mathcal{G}_{vi}) + \mathcal{L}_{min}(\mathcal{G}_{ir}, \mathcal{F}_{vi}) + \mathcal{L}_{min}(\mathcal{G}_{vi}, \mathcal{F}_{ir})$$
$$+ \mathcal{L}_{max}(\mathcal{G}_{ir}, \mathcal{F}_{ir}) + \mathcal{L}_{max}(\mathcal{G}_{vi}, \mathcal{F}_{vi}) \tag{17}$$

where $\mathcal{F}_{ir}$ and $\mathcal{F}_{vi}$ are the flattened $\Phi_{(ir)}$ and $\Phi_{(vi)}$, $\mathcal{L}_{min}(\cdot, \cdot)$ encourages feature dissimilarity between different modalities, and $\mathcal{L}_{max}(\cdot, \cdot)$ promotes similarity within the same modality.

# 4 EXPERIMENT

## 4.1 EXPERIMENTAL DETAIL

To evaluate our proposed M2PN, we conducted comprehensive experiments across four datasets: M³FD (Liu et al., 2023), ADD (Ahn et al., 2023), TNO (Toet, 2017), and MSRS (Tang et al., 2022a). These datasets were utilized for different tasks: all four datasets were employed for IVIF, while M³FD was additionally used for object detection and MSRS for image segmentation. Our M2PN was trained and validated on the RoadScene (Xu et al., 2020) dataset, then directly tested on the four test datasets to demonstrate its robustness and generalization capability. We compared M2PN against nine SOTA methods, including A²RNet (Li et al., 2025b), DCEvo (Liu et al., 2025a), FreeFusion (Zhao et al., 2025), GIFNet (Cheng et al., 2025), PromptFusion (Liu et al., 2024a), Text-IF (Yi et al., 2024), LRRNet (Li et al., 2023), CDDFuse (Zhao et al., 2023), and SHIP (Zheng et al., 2024). To ensure fair comparison, all models were obtained from their respective authors, and all experiments were implemented using PyTorch.

Table 1: Quantitative comparison results on the TNO, MSRS, M³FD and ADD. The best results are highlighted in **bold**.

| Method | TNO | | | | | | ADD | | | | | | MSRS | | | | | | M³FD | | | | | |
|---|---|---|---|---|---|---|---|---|---|---|---|---|---|---|---|---|---|---|---|---|---|---|---|---|
| | EN | SF | SD | AG | NI | VI | EN | SF | SD | AG | NI | VI | EN | SF | SD | AG | NI | VI | EN | SF | SD | AG | NI | VI |
| A²RNet | 7.05 | 3.43 | 9.47 | 3.29 | 4.75 | 0.56 | 6.28 | 2.76 | 9.41 | 2.11 | 3.77 | 0.39 | 6.60 | 3.49 | 8.56 | 2.92 | 4.28 | 0.66 | 6.60 | 3.49 | 8.56 | 2.92 | 4.28 | 0.66 |
| DCEvo | 6.91 | 4.01 | 9.29 | 3.94 | 4.49 | 0.44 | 6.48 | 4.31 | 9.32 | 3.24 | 3.99 | 0.56 | 6.64 | 4.52 | 8.36 | 3.81 | 4.45 | 0.83 | 6.64 | 4.52 | 8.36 | 3.81 | 4.45 | 0.83 |
| FreeFusion | 7.05 | 6.17 | 9.68 | 6.19 | 4.83 | 1.02 | 6.83 | 6.08 | 10.05 | 5.16 | 4.57 | **1.16** | 5.16 | 5.33 | 6.97 | 3.74 | 3.53 | 1.05 | 7.25 | 7.57 | 9.70 | 6.95 | 5.04 | 1.02 |
| GIFNet | 6.94 | 5.17 | 8.94 | 4.97 | 4.59 | 0.65 | 6.81 | **6.39** | 9.69 | 5.05 | 4.39 | 0.96 | 5.96 | 5.00 | 6.77 | 3.50 | 3.59 | 0.68 | 7.04 | 7.61 | 9.23 | 6.11 | 4.78 | 0.83 |
| PromptFusion | 7.01 | 4.22 | 9.20 | 4.17 | 4.67 | 0.55 | 6.60 | 4.21 | 9.12 | 3.00 | 4.10 | 0.65 | 6.65 | 4.36 | 8.33 | 3.61 | 4.35 | 0.79 | 6.78 | 5.31 | 10.03 | 4.46 | 4.43 | 0.47 |
| Text-IF | 7.21 | 5.18 | 9.54 | 5.17 | 4.89 | 0.70 | 6.99 | 5.23 | 9.48 | 4.32 | 4.70 | 1.01 | 6.74 | 4.67 | 8.52 | 3.95 | 4.49 | 0.91 | 6.93 | 6.21 | 9.88 | 5.34 | 4.66 | 0.63 |
| LRRNet | 7.05 | 3.81 | 9.17 | 3.86 | 4.64 | 0.46 | 6.75 | 3.82 | 9.49 | 3.22 | 4.26 | 0.63 | 6.19 | 3.31 | 7.82 | 2.67 | 3.74 | 0.43 | 6.44 | 4.21 | 9.31 | 3.61 | 4.05 | 0.89 |
| CDDFuse | 7.09 | 4.55 | 9.38 | 4.51 | 4.76 | 0.61 | 6.65 | 4.62 | 9.24 | 3.31 | 4.21 | 0.76 | 6.71 | 4.51 | 8.43 | 3.77 | 4.49 | 0.83 | 6.90 | 5.77 | 9.97 | 4.81 | 4.66 | 0.89 |
| SHIP | 6.93 | 4.76 | 9.25 | 4.69 | 4.47 | 0.40 | 6.49 | 4.89 | 9.12 | 4.05 | 3.96 | 0.58 | 6.44 | 4.64 | 8.15 | 3.97 | 4.17 | 0.79 | 6.83 | 6.03 | 10.01 | 5.20 | 4.49 | 0.90 |
| **Ours** | **7.34** | **6.78** | **9.64** | **6.88** | **5.14** | **1.20** | **7.33** | 5.86 | **10.60** | **5.20** | **5.06** | 1.09 | **7.16** | **6.76** | **9.18** | **6.53** | **4.87** | **1.09** | **7.36** | **8.28** | **10.21** | **7.17** | **5.15** | **1.23** |

Table 2: Performance on high-level vision tasks. The best results are highlighted in **bold**.

| Method | Object Detection | | | | | | | Semantic Segmentation | | | | | | |
|---|---|---|---|---|---|---|---|---|---|---|---|---|---|---|
| | Per | Car | Bus | Mot | Tru | Lam | mAP | UnL | Car | Per | Bike | Cur | Stop | mIoU |
| A2RNet | 0.795 | 0.906 | 0.886 | 0.659 | 0.810 | 0.798 | 0.809 | 0.9818 | 0.8809 | 0.6875 | 0.6882 | 0.5557 | 0.6445 | 0.74 |
| DCEvo | 0.780 | 0.907 | 0.891 | 0.675 | 0.788 | 0.815 | 0.809 | 0.9819 | 0.8775 | 0.6865 | 0.6792 | 0.5243 | 0.6383 | 0.731 |
| FreeFusion | 0.785 | 0.910 | 0.887 | 0.695 | 0.807 | 0.802 | 0.814 | 0.9794 | 0.8591 | 0.6808 | 0.6432 | 0.4516 | 0.5715 | 0.698 |
| GIFNet | 0.787 | 0.907 | 0.881 | 0.702 | 0.765 | 0.811 | 0.809 | 0.9814 | 0.8730 | 0.6862 | 0.6902 | 0.5437 | 0.6207 | 0.733 |
| PromptFusion | 0.775 | **0.911** | 0.890 | 0.648 | 0.813 | 0.806 | 0.807 | 0.9814 | 0.8655 | 0.6744 | 0.6809 | 0.5388 | 0.6514 | 0.732 |
| Text-IF | 0.773 | 0.907 | **0.905** | 0.693 | 0.810 | 0.795 | 0.814 | 0.9821 | 0.8798 | 0.6829 | 0.6985 | 0.5355 | 0.6373 | 0.736 |
| LRRNet | 0.780 | **0.911** | 0.878 | 0.694 | 0.804 | 0.798 | 0.811 | 0.9817 | 0.8800 | 0.6762 | 0.6787 | 0.5475 | 0.6417 | 0.734 |
| CDDFuse | 0.788 | **0.911** | 0.888 | 0.698 | **0.820** | 0.789 | 0.816 | 0.9818 | 0.8717 | 0.6911 | 0.6891 | 0.5578 | 0.6569 | 0.741 |
| SHIP | 0.790 | 0.909 | 0.877 | 0.673 | 0.812 | 0.810 | 0.812 | 0.9824 | **0.8862** | 0.6961 | 0.6918 | **0.5662** | 0.6307 | 0.742 |
| **Ours** | **0.794** | 0.910 | 0.880 | **0.712** | 0.786 | **0.824** | **0.818** | **0.9825** | 0.8814 | **0.7064** | **0.6999** | 0.5487 | **0.6647** | **0.747** |

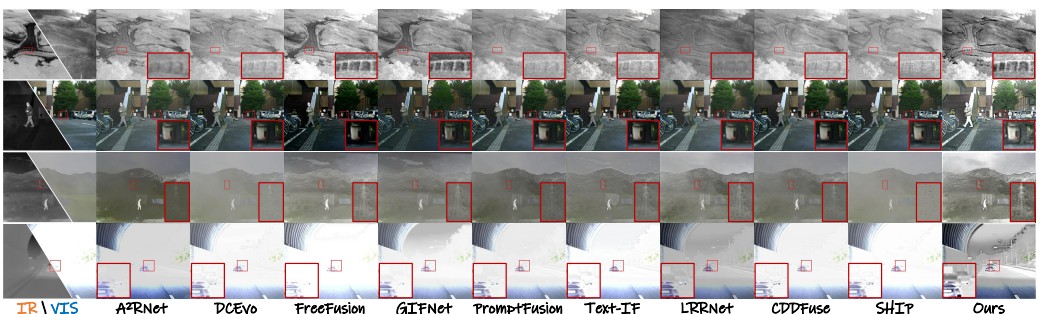

Figure 4: Visual comparison of different methods on the TNO, MSRS, M³FD and ADD.

## 4.2 FUSION RESULT

We employed six quantitative quality assessment metrics to evaluate the fusion results: Entropy (EN), Spatial Frequency (SF), Standard Deviation (SD), Average Gradient (AG), Nonlinear Information Quantity-based Metric (NI), and Visual Information Fidelity in Frequency domain (VI). For all these metrics, higher values indicate superior fusion performance (Liu et al., 2024b; Zhang & Demiris, 2023). **Qualitative comparisons:** Figure 4 presents a qualitative comparison between our M2PN and SOTA methods. It is evident that our method excels in preserving textural details and thermal radiation information, particularly in challenging scenarios such as trees in darkness, signal towers in fog, and vehicles in overexposed regions. These improvements facilitate a better understanding of complex scenes. **Quantitative Comparisons:** Subsequently, we conducted quantitative comparisons using six evaluation metrics, as shown in Table 1. Our method demonstrates superior performance across nearly all metrics, validating that our method effectively integrates complementary features from cross-modal inputs. This integration enables the fused images to achieve higher fidelity, preserve more edge information, and exhibit reduced distortion.

## 4.3 PERFORMANCE IN HIGH-LEVEL VISION TASKS

We evaluate the proposed method on the M³FD and MSRS datasets for object detection and semantic segmentation, with results summarized in Table 2. Our method achieves the highest mAP and

Figure 5: A visualization of the ablation experiment.

mIoU across both tasks, confirming its robustness and generalization. Specifically, it delivers clear improvements in pedestrian (Per), motorcycle (Mot), and lamp post (Lam) detection, as well as in unlabeled (UnL), pedestrian (Per), bicycle (Bike), and car stop (Stop) segmentation, showing advantages in handling small objects and complex semantic regions. These gains arise from the synergy between memory-driven experience prompts and modality-specific prompts. The former stabilizes feature learning by reusing high-quality fusion patterns, while the latter provides modality-aware guidance, enabling fused images with stronger structural consistency and semantic separability. As a result, our design enhances both statistical metrics and high-level vision performance.

## 4.4 ABLATION STUDY

We conducted nine ablation experiments to systematically evaluate our proposed method, with qualitative and quantitative results shown in Figure 5 and Table 3, respectively. **Cases 1-3 investigate core components**: Case 1 (w/o $\Theta_{\mathcal{M}}\&\Theta_f$) removes residual structure and fusion priors to assess CSGN's fine-grained modality perception; Case 2 ($CLIP \rightarrow \mathbb{D}$) replaces CLIP with a discriminator following Wang et al. (2025b) to evaluate text priors' effectiveness; Case 3 (w/o CLIP) removes CLIP evaluation in DMB, using self-updating instead. **Cases 4-5 examine module contributions**: Case 4 (w/o CSGN) and Case 5 (w/o DMB) directly remove CMGN and DMB re-

Table 3: Ablation study results on validation dataset. The best results are highlighted in **bold**.

| Method | EN | SF | SD | AG | NI | VI |
|---|---|---|---|---|---|---|
| w/o $\Theta_{\mathcal{M}}\&\Theta_f$ | 7.16 | 6.22 | 10.09 | 5.95 | 4.71 | 0.50 |
| $CLIP \rightarrow \mathbb{D}$ | 7.36 | 6.98 | 10.24 | 6.69 | 5.04 | 0.64 |
| w/o $CLIP$ | 7.38 | 6.94 | 10.19 | 6.74 | 5.05 | 0.65 |
| w/o $CSGN$ | 7.21 | 6.21 | 10.08 | 5.96 | 4.79 | 0.52 |
| w/o $DMB$ | 7.20 | 6.30 | 10.08 | 5.99 | 4.78 | 0.51 |
| w/o $\mathcal{L}_s$ | 7.08 | 5.07 | **10.58** | 5.07 | 4.83 | 0.38 |
| w/o $w$ | 7.25 | 6.85 | 10.16 | 6.56 | 4.87 | 0.58 |
| w/o $\mathcal{L}_{ctr}$ | 7.23 | 6.47 | 10.13 | 6.23 | 4.82 | 0.55 |
| w/o $\Theta_p$ | 7.26 | 6.42 | 10.04 | 6.17 | 4.87 | 0.57 |
| **Ours** | **7.53** | **8.25** | 10.38 | **7.94** | **5.30** | **0.88** |

spectively to verify their effectiveness in prompt-based learning. **Cases 6-8 analyze loss function components**: removing regularization term $\mathcal{L}_s$ (Case 6), adaptive weight $w$ (Case 7), and contrastive learning $\mathcal{L}_{ctr}$ (Case 8). **Case 9** (w/o $\Theta_p$) removes all prompt learning components to validate their contribution to model optimization. Results demonstrate that our dual-prompt guided method achieves superior performance through effective collaboration among all modules.

## 5 CONCLUSION

This study proposes the M2PN model, which transforms image fusion from a static feature aggregation process into a dynamic prompt-guided learning paradigm. By introducing cross-modal prompts and a memory mechanism, the model achieves efficient modeling and dynamic balancing of different modality features, thereby enhancing detail preservation and semantic consistency in the fusion process. In addition, specifically designed modules such as CSGN and DMB ensure the effective output of prompts. Furthermore, by quantifying information contributions and adaptively assigning weights, the model narrows the solution space and better preserves source image features. Experimental results demonstrate that M2PN not only outperforms existing methods in quantitative metrics but also exhibits stronger robustness and generalization in complex environments, effectively facilitating the deployment of high-level vision tasks.

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

# A APPENDIX

## A.1 MOTIVATION REVIEW

Contemporary IVIF methods suffer from a fundamental limitation: they treat each fusion instance as an isolated optimization problem, discarding valuable knowledge from successful fusion experiences. Consider the conventional fusion formulation:

$$fused = \mathbb{N}(IR, VIS; \Psi) \tag{18}$$

where $\mathbb{N}$ represents the fusion function with static parameters $\Psi$, ignoring historical success patterns.

Human visual perception excels at fusion tasks by leveraging experiential knowledge—unconsciously drawing upon patterns from previous similar scenarios. This cognitive mechanism motivates a dynamic learning paradigm that accumulates and utilizes historical fusion experiences:

$$fused = \mathbb{N}(IR, VIS, \mathbb{B}; \Psi) \tag{19}$$

where $\mathbb{B}$ represents accumulated experiential knowledge.

However, experiential knowledge alone is insufficient due to fundamental modality heterogeneity. Infrared images capture thermal radiation patterns while visible images provide textural details under favorable illumination—resulting in distinct information distributions and semantic expressions. Conventional approaches apply uniform processing strategies, failing to capitalize on each modality's unique advantages and potentially introducing harmful cross-modal interference.

The challenge extends beyond separate processing to understanding complex interdependencies and complementary relationships. Cross-modal semantic relationships are inherently non-linear and context-dependent, requiring sophisticated modeling that captures both intra-modal characteristics and inter-modal interactions simultaneously. Traditional linear combinations or simple attention mechanisms are insufficient for these complex dynamics.

Graph-based representations offer a natural solution for modeling such relational structures. Unlike rigid sequential or convolutional architectures, graph networks flexibly represent arbitrary relationships and enable information propagation along semantically meaningful pathways. This capability is particularly valuable for modeling intricate dependencies between modalities and their fusion outcomes, as graphs can encode relationships as edges while representing modal features as nodes.

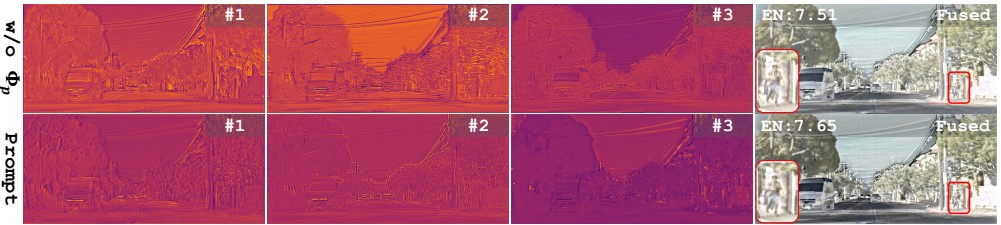

Figure 6: Visualization of features before and after prompt learning.

Therefore, our design motivation centers on a dual-prompt learning framework combining historical experiential knowledge for global guidance with graph-based cross-modal reasoning for modality-specific understanding:

$$fused = \mathbb{N}(IR, VIS, \mathbb{B}, \mathcal{G}; \Psi) \tag{20}$$

where $\mathcal{G}$ represents graph-based cross-modal semantic reasoning, addressing both temporal learning from experience and structural modeling of complex cross-modal relationships for sophisticated adaptive fusion decisions.

Figure 6 presents the visualization of feature maps and fused images before and after prompt injection. It is evident that with the introduction of prompts, the model is better able to capture complementary feature representations from the source features, particularly enhancing the preservation of texture and salient features. Consequently, the fused images exhibit more discriminative feature representations, thereby improving their information content.

## A.2 CLIP-BASED QUALITY EVALUATION

The fundamental challenge in unsupervised image fusion lies in establishing reliable quality assessment criteria without ground truth references. Traditional metrics such as entropy, mutual information, or gradient-based measures often fail to capture perceptual quality that aligns with human visual perception. To address this limitation, we leverage the robust cross-modal understanding capabilities of CLIP (Contrastive Language-Image Pre-training) to construct a semantically-aware quality evaluator that can assess fusion results from a human-centric perspective.

The motivation for employing CLIP stems from three key observations: i) CLIP's large-scale pre-training on diverse image-text pairs enables it to understand high-level semantic concepts of image quality; ii) Its contrastive learning paradigm naturally supports comparative quality assessment through similarity computation; iii) The text-guided evaluation provides interpretable quality criteria that can be explicitly defined and adjusted.

**Quality Assessment Framework.** Our CLIP-based evaluator operates through a contrastive categories of text-image matching paradigm. We design two complementary textual descriptions that capture the essential characteristics of high-quality and low-quality fusion results:

---

**Algorithm 1** Training of the proposed M2PN

---

**Input:** $IR\&VIS$

**Random initialization:** Siamese-DenseEncoder: $\mathbb{E}_{SD}(\cdot)$, Residual Encoder: $\mathbb{E}_R(\cdot)$, $CSGN(\cdot)$, $DMB(\cdot)$, $PGM(\cdot)$

**Fixed Parameters:** $\lambda = 15$; Training epoch: $K$; Batch size: 16; Initial learning rate: 0.001

---

1: **for** $n = 1$ **to** $K_p$ **do**
2:   **while** not complete all iterations **do**
3:     *% Feature Extraction*
       $\Phi_{ir}, \Phi_{vi} \leftarrow \mathbb{E}_{SD}(IR, VIS); \Theta_{\mathcal{M}} \leftarrow \mathbb{E}_R(IR - VIS)$
4:     $\Phi_f \leftarrow Cov(CAT(\Phi_{ir}, \Phi_{vi}))$
5:     $\mathcal{G}_{ir} \leftarrow CSGM(MLP(\Phi_{ir}), MLP(\Phi_f), \Theta_f)$
       $\mathcal{G}_{vi} \leftarrow CSGM(MLP(\Phi_{vi}), MLP(\Phi_f), \Theta_f)$
       *% Generating modality-specific cues by building a graph of cross-modal features*
6:     $\Theta_q \leftarrow DMB(\Theta_f)$ *% Query the memory bank to provide current fusion clues*
7:     **for** $it = 1$ **to** 3 **do**
8:       *% Stepwise integration of fusion prompts for feature reconstruction guidance*
         $\Phi_f = PGM(\Phi_f; MGF(\mathcal{G}_{ir}, \mathcal{G}_{vi}, \Theta_q))$
9:     **end for**
10:    $fused = Tanh(\Phi_f)$; *% Utilize $\mathcal{L}_{total} \leftarrow Eq.(10, 17)$ to update the all model*
11:    $Quality \leftarrow CLIP(Fused)$
       *% Employ positive and negative textual descriptors based on image texture, contrast, and luminance characteristics to evaluate the quality of fused images and dynamically update the memory bank with the assessment results*
12:  **end while**
13: **end for**

---

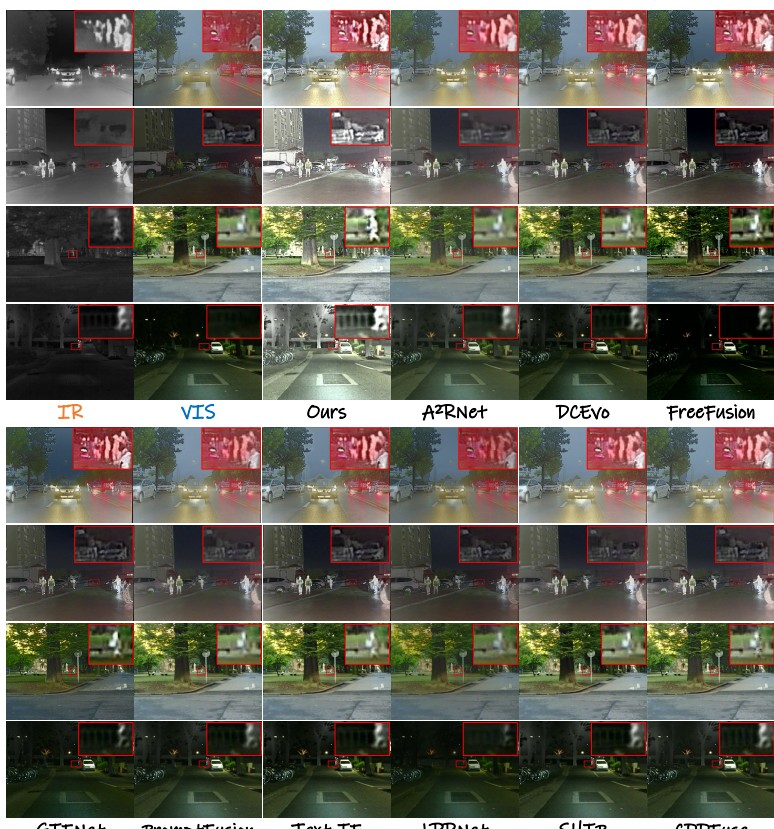

Figure 7: Visualization of the high-level vision task results.

Positive Quality Prompt ($T_{pso}$):*"A high-quality image with clear contrast, sharp details, proper brightness, clean composition without noise or artifacts."*; *"An excellent image showing sharp details, accurate tones, optimal lighting, and no noise or artifacts."*; *"A clear, well-defined image with precise textures, natural brightness, and flawless composition without imperfections."*

Negative Quality Prompt ($T_{neg}$):*"Low-quality image with poor contrast, blurry details, improper brightness, significant noise and visible artifacts."*; *"An excellent image showing sharp details, accurate tones, optimal lighting, and no noise or artifacts."*,*"A clear, well-defined image with precise textures, natural brightness, and flawless composition without imperfections."*; *"An unclear image with blurred edges, poor exposure, and visible grain or compression artifacts."*; *"Distorted image with dull contrast, missing details, uneven brightness, and distracting noise patterns."*

These prompts encapsulate multiple dimensions of perceptual quality including contrast preservation, detail clarity, brightness appropriateness, compositional coherence, and artifact suppression, all critical aspects for evaluating fusion effectiveness.

## A.3 EXPERIMENT SETUP

**Implementation Details:** Our M2PN is implemented on a single NVIDIA 2080Ti GPU with 11 GB memory, running at 3.0 GHz with an Intel i7-9700 CPU. We employ the Adam optimizer with a batch size of 16 and an initial learning rate of 0.001, utilizing thermal decay for model training. To align the input data modalities, we utilize the YCbCr color space to separate the luminance and chrominance components of $VIS$, and restore the $VIS$ chrominance of the fused image after fusion. The overall training strategy can be found in Algorithm 1.

**Benchmark Datasets:** We randomly crop the RoadScene dataset into 8,000 pairs of $128 \times 128$ patches for training, with 40 image pairs selected as the validation set. We randomly select 40 image

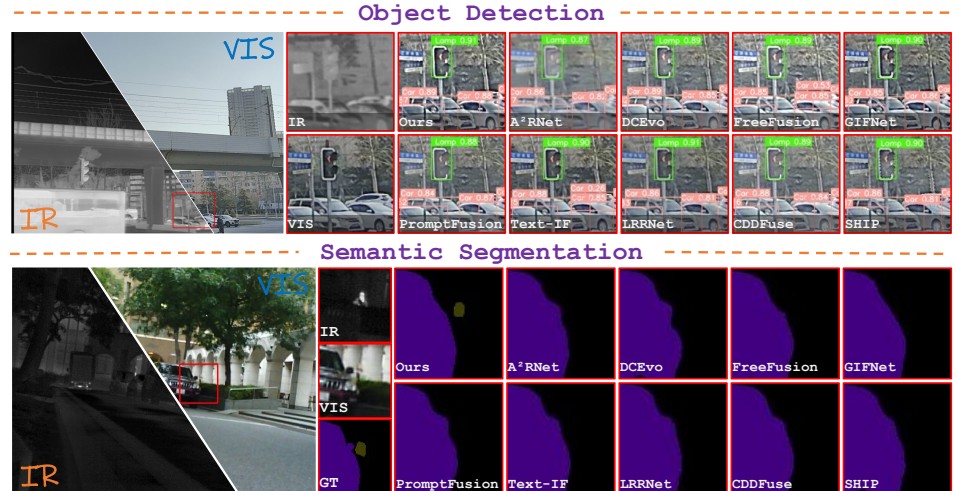

Figure 8: More visualizations of fused results on the test dataset.

pairs from the TNO dataset for testing. For the M³FD dataset, we adopt the *"independent scene for fusion"* subset as the test dataset. Regarding the MSRS dataset, we randomly select 361 image pairs as the test dataset. We construct a test dataset of 24 scene pairs by randomly sampling frames from the ADD sequence dataset.

For the object detection task, we utilize the *"for fusion, detection and fused-based detection"* subset from M³FD, constructing training, validation, and test sets in a 6:2:2 ratio, and select YOLOv5 as the detector.

In the semantic segmentation task, we used the training set provided by MSRS to retrain the segmentation network (Cao et al., 2023) to explore the performance of M2PN.

### A.4 MORE RESULTS

**Additional Results on test dataset:** Figure 7 presents comprehensive fusion results, clearly demonstrating that our method achieves superior visual performance, particularly excelling in small target detection. Notably, our M2PN effectively emphasizes the source feature information, with this enhancement becoming more pronounced in nighttime scenarios. We attribute this phenomenon primarily to our text prompts that specifically emphasize texture, contrast, and brightness. Consequently, after coupling modal-specific representations, these prompts effectively drive the model to highlight crucial source feature information. In comparison, Text-IF, despite being driven by prompt learning, exhibits relatively poor performance due to limitations inherent in CLIP's knowledge structure. While PromptFusion employs learnable prompts to better adapt to open environments, it lacks modal-specific self-prompting mechanisms. This deficiency results in mutual suppression between modal feature representations, leading to conflicting performance outcomes.

**High-level Vision Task:** Figure 8 demonstrates the qualitative results of our proposed method on object detection and semantic segmentation tasks. Guided by historical experience and modality-specific information, our M2PN can effectively identify and perceive salient representations within source features. Consequently, it provides fine-grained scene information for high-level vision tasks, maintaining robust performance even in challenging scenarios involving small targets or dense regions, such as vehicle detection and the detection of pedestrians behind vehicles—capabilities that other methods struggle to achieve.

**CMGN Performance Visualization:** To validate the effectiveness of CMSG components, we conducted comprehensive ablation experiments. As illustrated in Figure 9, the t-SNE visualization demonstrates the facilitating effects of components $\Theta_f \& \Theta_{\mathcal{M}}$ on modality-specific learning. The experimental results reveal that components $\Theta_f \& \Theta_{\mathcal{M}}$ significantly enhance the model's capacity

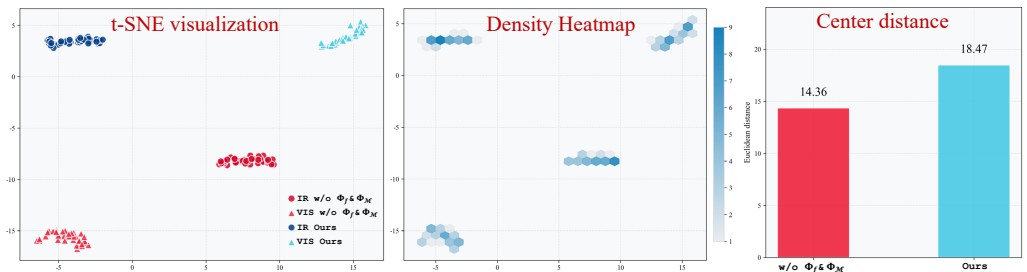

Figure 9: Visualization of features before and after prompt learning.

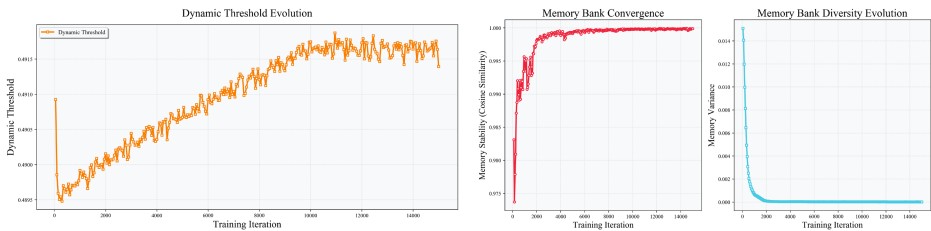

Figure 10: Visualization of dynamic thresholds and status of the DMB.

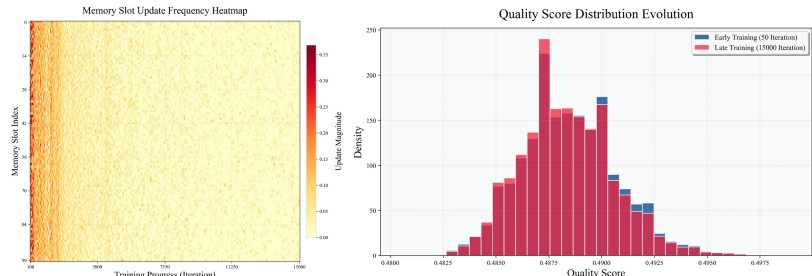

Figure 11: Visualization of DMB internal feature responses before and after training.

to perceive distinct modality attributes, resulting in more compact intra-class clustering and more pronounced inter-class separation in the visualization space, thereby confirming the efficacy of the proposed M2PN.

**DMB Performance Visualization:** Regarding the design effectiveness of the DMB, we conducted systematic ablation studies to investigate its validity. Figures 10 and 11 present the complete learning evolution process of DMB features. Experimental observations reveal that the designed memory bank mechanism exhibits a beneficial evolutionary trajectory from unstable to stable states throughout the training process: (i) The dynamic threshold demonstrates progressive convergence characteristics during iterative training, ensuring the rationality and consistency of the sample selection strategy; (ii) The similarity metrics within the memory bank rapidly improve and eventually stabilize, reflecting the mechanism's favorable convergence properties;

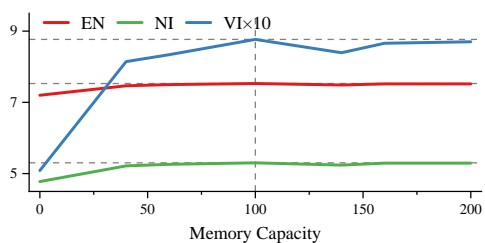

Figure 12: Effect of memory bank capacity on model performance.

(iii) Feature diversity gradually decreases as training progresses, forming more compact and discriminative feature representations; (iiii) The slot update frequency transitions from frequent adjustments in early training phases to balanced fine-tuning in later stages, while the quality distribution evolves

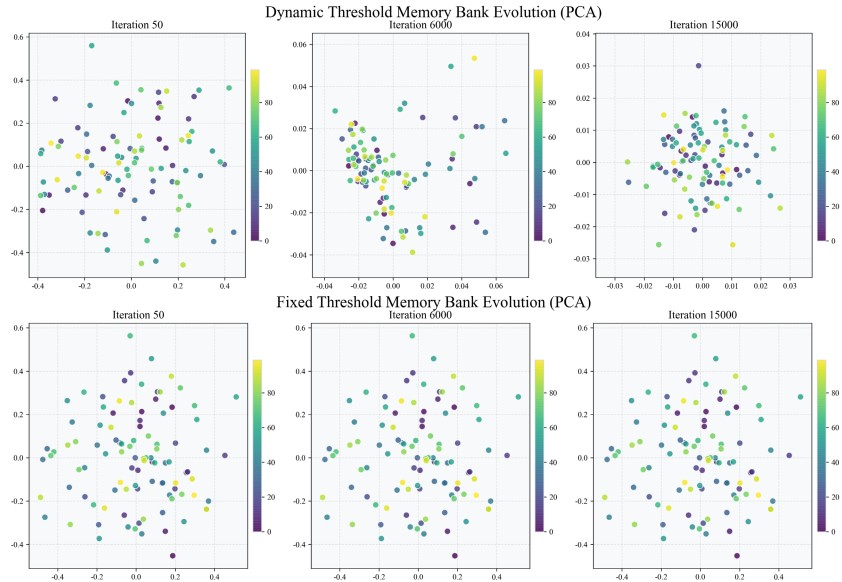

Figure 13: Visualization of the effectiveness of dynamic thresholding on DMB.

Table 4: Comparison with different text prompts. The best results are highlighted in **bold**.

| Method | EN | SF | SD | AG | NI | VI |
|---|---|---|---|---|---|---|
| *Definition of fusion.* | 7.522 | 8.104 | **10.395** | 7.758 | 5.298 | 0.868 |
| *Text prompt for Text-IF.* | 7.525 | 8.146 | 10.392 | 7.804 | 5.297 | 0.864 |
| **Ours** | **7.530** | **8.249** | 10.381 | **7.941** | **5.303** | **0.877** |

from an initially scattered state to convergence within high-quality intervals. These experimental findings collectively demonstrate that the proposed mechanism can effectively establish a stable and high-quality feature storage system. In Figure 12, we further examine the effect of memory bank capacity on the model. Intuitively, setting the capacity to 100 yields near-optimal performance, and larger capacities only fluctuate around this optimum. Therefore, considering efficiency and performance trade-offs, we set the memory bank capacity to 100.

Furthermore, we specifically investigated the effectiveness of the dynamic threshold design. As shown in Figure 13, the experimental results demonstrate that introducing the dynamic threshold strategy significantly optimizes the evolutionary trajectory of the memory bank, enabling faster convergence to stable states during training while facilitating the storage of more discriminative, high-quality feature representations. This further validates both the necessity and effectiveness of the proposed design.

**Text prompt analysis.** To evaluate the impact of fusion text versus quality text on model performance, we

Table 5: Computational efficiency of SOTA methods on the validation dataset. The best results are highlighted in **bold**.

| Method | Para. (M) | FLOPS (G) | FPS | EN |
|---|---|---|---|---|
| A$^2$RNet | 10.61 | 36.5 | 0.16 | 7.30 |
| DCEvo | 2.01 | 195 | 1.43 | 7.18 |
| FreeFusion | 5.67 | 96.7 | 6.54 | 7.12 |
| GIFNet | 0.82 | 39.0 | 2.10 | 7.35 |
| PromptFusion | 7.78 | - | 3.15 | 7.41 |
| Text-IF | 336.8 | 215 | 2.78 | 7.38 |
| LRRNet | **0.05** | **3.3** | 8.19 | 7.14 |
| CDDFuse | 1.19 | 32.8 | 3.60 | 7.45 |
| SHIP | 0.55 | 35.2 | 2.08 | 7.16 |
| Ours | 0.97 | 15.6 | **25.6** | **7.53** |

conducted comparative experiments, with results presented in Table 4. We employed a typical fusion definition text: *"This image effectively integrates the thermal radiation information from the*

*infrared image and the texture details from the visible light image,"* and compared it with the fusion text used by Text-IF: *"This is the infrared and visible light image fusion task."* As discussed in the previous section, fusion definitions are relatively abstract for CLIP models due to their inherent knowledge structure. In contrast, our approach can more effectively guide the model to generate high-quality images by explicitly defining fusion quality criteria.

**Performance on the validation dataset:**   We conducted a comprehensive evaluation of M2PN's computational efficiency on the validation dataset, encompassing key metrics including learnable parameters (Para), floating-point operations (FLOPs) at $256 \times 256$ resolution, frames per second (FPS), and EN. As presented in Table 5, our method demonstrates a compelling trade-off between computational efficiency and performance quality. While M2PN exhibits higher parameter count and FLOPs compared to the algorithm-unrolling based LRRNet, it achieves superior FPS and EN scores, establishing a solid foundation for practical deployment. Moreover, when compared to text-prompt-based approaches such as Text-IF and PromptFusion, our method leverages historical experience and modality-specific prompt learning to achieve enhanced performance with significantly reduced computational overhead. This efficiency gain validates the effectiveness of our design philosophy, which prioritizes intelligent prompt construction over brute-force parameter scaling.

ACKNOWLEDGMENTS

The authors acknowledge the assistance of **Claude** in the preparation of this manuscript. **Claude** was utilized specifically for language editing, including grammar correction, logical structure refinement, and proofreading. All scientific content, methodology, analysis, and conclusions remain entirely the work of the authors.

