# OpenReview forum: "Memory-Orchestrated Multi-Prompt Learning for Infrared and Visible Image Fusion"
_ICLR.cc/2026/Conference — ICLR 2026 Conference Withdrawn Submission_

### Official Review · Reviewer_M4Rc · 2025-10-21

**Soundness:** 3
**Presentation:** 2
**Contribution:** 3
**Rating:** 4
**Confidence:** 4

**Summary:**

To address the difficulty of leveraging historical fusion experience and the lack of modality-specific semantic guidance in existing methods, this work proposes a memory-coordinated multi-prompt learning network. The algorithm uses CLIP to evaluate fusion quality and store the results in a memory bank, and employs a graph-structured model to capture cross-modal semantic information. Extensive experiments validate the effectiveness of the proposed approach.

**Strengths:**

1. This work transforms the fusion process from static feature combination to a dynamic, prompt-driven learning paradigm, effectively improving fusion performance.
2. It builds a self-evolving dynamic memory bank with CLIP, which stores high-quality fused feature representations from historical learning scenarios.
3. The experiments are comprehensive, with effectiveness evaluated on four image-fusion datasets and two downstream tasks.

**Weaknesses:**

1. The proposed algorithm quantifies explicitly defined quality attributes—such as texture, contrast, and brightness—via CLIP’s evaluation capability. Consequently, fusion performance depends, to some extent, on CLIP’s ability to assess multimodal information, and the constructed memory bank may contain errors. In addition, in Section A.2, the negative prompt list includes sentences similar to the positive ones, leading to contradictions within the text set.
2. “CSGN” is repeatedly written as “CSGM/CMGN/CMSG” in the algorithm and main text, and Equation (8) explains Θ_p as Θ_q. Such low-level mistakes recur in the methods section, reducing readability and raising concerns about the reliability of the proposed approach.
3. The mathematical derivations are not coherent, with large jumps that make it difficult for readers to grasp the core ideas. Moreover, the definitions and settings of different parameters are not fully specified.
4. The introduction mentions the third challenge of prompt learning-based IVIF methods, but the manuscript does not provide a clear solution. How to “translate human-perceived quality into learnable constraints” requires a deeper discussion.
5. The comparative experiments are not fair. Many existing algorithms are trained on datasets such as MSRS or M3FD and rely heavily on the visible modality to ensure fidelity of fusion results. Since ADD contains degraded visible images (e.g., overexposure), all compared methods should be retrained on this dataset for a fair comparison.

**Questions:**

1. Can the proposed algorithm generalize to more complex scenarios, such as adverse weather and noise? Compared with algorithms specifically designed for these scenarios, would it achieve performance gains?
2. The algorithm is trained on RoadScene and tested on the remaining datasets. Why not choose LLVIP or M3FD as the training dataset?

---

### Official Review · Reviewer_Hu7s · 2025-10-28

**Soundness:** 3
**Presentation:** 3
**Contribution:** 2
**Rating:** 4
**Confidence:** 4

**Summary:**

This paper proposes M2PN (Memory-Orchestrated Multi-Prompt Learning Network) for infrared and visible image fusion (IVIF).The main idea is to transform static feature combination into a dynamic, prompt-guided paradigm by introducing two modules: Dynamic Memory Bank (DMB) and Cross-Modal Semantic Graph Network (CSGN). These dual prompts guide the fusion process progressively, while residual priors constrain the fusion space.

**Strengths:**

+ The design of CLIP-evaluated dynamic memory and graph-based prompt generation is coherent and systematically integrated into the network pipeline.
+ Evaluation across multiple benchmarks and high-level vision tasks demonstrates both low-level fusion quality and high-level transferability.
+ The paper is clearly written, logically organized, and includes detailed motivation, figures, and pseudo-code.

**Weaknesses:**

- Using CLIP similarity as a quality evaluator introduces heuristic choices (positive/negative prompts, scaling factor). The reliability and reproducibility of this component should be further validated.
- The claim that CLIP can act as a “semantic quality judge” for IVIF results is questionable because CLIP was trained on RGB natural images, not infrared or fused data.
- Although the memory module is central to the paper, there is limited analysis of how the stored experiences generalize to unseen cases.
- The proposed M2PN introduces multiple heavy modules (CSGN, DMB, multi-stage prompt injection). The paper lacks runtime or computational cost analysis, which is important for real-time IVIF tasks.
- The fusion loss combines several heuristic weighting mechanisms (entropy-based, residual-based). The derivation could be more principled or supported by sensitivity studies.
- The ablations remove components but do not analyze parameter sensitivity (e.g., memory capacity N, update rate β, prompt scales). Such analysis would strengthen robustness claims.

**Questions:**

Please refer to the Weaknesses.

---

### Official Review · Reviewer_vf48 · 2025-11-01

**Soundness:** 3
**Presentation:** 3
**Contribution:** 3
**Rating:** 4
**Confidence:** 4

**Summary:**

This paper proposes a Memory-Orchestrated Multi-Prompt Network (M2PN) for infrared–visible image fusion. The key idea is to unify experience-driven prompting and modality-specific prompting under a single framework. A Dynamic Memory Bank (DMB) stores high-quality historical fusion representations and retrieves “experience prompts” via quality-gated writing; a Cross-modal Semantic Graph Network (CSGN) builds a three-node semantic graph for each modality to generate modality-aware prompts; and a Prompt Guidance Module (PGM) injects prompts layer-by-layer during decoding, combined with a Memory-Guided Fusion (MGF) and AdaIN-based conditional modulation. The loss includes weighted fidelity and structure/texture preservation terms, as well as contrastive constraints. The method is evaluated on classical fusion benchmarks and downstream tasks (detection, segmentation), and ablation studies verify the role of memory and graph prompts.

**Strengths:**

1.Paradigm innovation. The paper integrates “experience memory” and “modality-aware prompts” into a unified prompting-guided fusion pipeline, addressing the lack of dynamic guidance and modality-discriminative supervision in prior fusion networks.
2.Systematic design. The pipeline forms a complete reasoning loop: initial fused features → semantic graph → memory retrieval → prompt modulation → adaptive decoding. The design is logically consistent.
3.Comprehensive experiments. Evaluations include classical fusion metrics, downstream detection/segmentation tasks, and ablations demonstrating the benefit of memory and graph prompts.

**Weaknesses:**

1.Quality gating depends on CLIP-based scoring, risking semantic bias and memory contamination.
The Dynamic Memory Bank writes entries only when the CLIP-derived quality score exceeds a dynamic threshold, then updates the most similar slot via momentum. However, quality scoring relies on CLIP similarity to positive/negative text prompts, which may not faithfully capture infrared fidelity, thermal edges, or cross-modal geometric consistency. Without multi-source validation or contamination control, the memory could accumulate biased or suboptimal samples, particularly under domain shifts.
2.Memory lifespan and stability are under-explored.
Memory updates only the most similar slot with a fixed momentum β, without discussing capacity management, replacement strategy, or forgetting. No experiments demonstrate long-term stability, catastrophic drift prevention, or behavior under continuous training/online deployment.
3.Restricted graph expressiveness.
CSGN builds a three-node fixed-structure graph per modality ({modality token, fusion token, memory token}). While simple, this topology may be insufficient to capture fine-grained cross-modal structural correspondences (e.g., thermal boundaries vs. texture edges). No comparisons to larger, hierarchical, or region-level graphs are reported, leaving unclear whether the graph design is optimal or merely functional.
4.Loss design contains heuristic elements; robustness not fully validated.
The weighted fidelity term uses dual-stage saliency and residual entropy heuristics to estimate modality importance. Robustness under noise, intensity non-uniformity, thermal drift, and IR artifacts (e.g., stripe noise) is not studied. Lack of stress-test evidence limits confidence in generalization to real infrared conditions.
5.Efficiency claims need more standardized evaluation.
FLOPs and FPS are reported on 256×256 input and compared to text-prompt-based fusion. However, comparisons to strong transformer-based fusion baselines under identical settings are limited, and memory footprint / latency curves on high-resolution inputs are absent. Deployment feasibility needs clearer quantitative support.

**Questions:**

1.Memory reliability:
How does the model prevent memory contamination when CLIP quality scoring is biased or fails under out-of-distribution thermal scenes? Could multi-metric or geometric-consistency-based gating be used?
2.Memory management:
What is the behavior when memory capacity increases? Have you evaluated alternative update policies (e.g., top-k slot update, diversity-preserving replacement, or decay-based forgetting)?
3.Graph expressiveness:
Why choose a fixed three-node graph? Would region-level or adaptive graph structures offer further gains? Have you measured cost–performance trade-offs?
4.Loss robustness:
How stable are the saliency and entropy-based weights under severe noise, illumination fluctuations, or real sensor artifacts? Any sensitivity curves or sim-to-real tests?

---

### Note · Authors · 2025-11-13

I have read and agree with the venue's withdrawal policy on behalf of myself and my co-authors.